# Quench Products of K-Ca-Mg Carbonate Melt at 3 and 6 GPa: Implications for Carbonatite Inclusions in Mantle Minerals

**Anton V. Arefiev [1],\*, Anton Shatskiy [2], Altyna Bekhtenova [1] and Konstantin D. Litasov [3]**

1   Sobolev Institute of Geology and Mineralogy, Siberian Branch of the Russian Academy of Sciences, 630090 Novosibirsk, Russia

2   Vernadsky Institute of Geochemistry and Analytical Chemistry, Russian Academy of Sciences, 119991 Moscow, Russia

3   Vereshchagin Institute of High Pressure Physics, Russian Academy of Sciences, Troitsk, 108840 Moscow, Russia

\*   Correspondence: arefievanton@igm.nsc.ru

**Abstract:** Alkali-rich carbonate melts are found as inclusions in magmatic minerals, mantle xenoliths, and diamonds from kimberlites and lamproites worldwide. However, the depth of their origin and bulk melt composition remains unclear. Here, we studied quench products of K-Ca-Mg carbonate melt at 3 and 6 GPa. The following carbonates were detected at 3 GPa: $K_2CO_3$, $K_2Ca(CO_3)_2$ bütschliite ($R\overline{3}2/m$), o-$K_2Ca_3(CO_3)_4$ ($P2_12_12_1$), $K_2Ca_2(CO_3)_3$ ($R3$), $K_2Mg(CO_3)_2$ ($R\overline{3}m$), Mg-bearing calcite, dolomite, and magnesite. At 6 GPa, the variety of quench carbonate phases includes $K_2CO_3$, $K_2Ca(CO_3)_2$ bütschliite ($R\overline{3}2/m$), d-$K_2Ca_3(CO_3)_4$ ($Pnam$), $K_2Mg(CO_3)_2$ ($R\overline{3}m$), aragonite, Mg-bearing calcite, dolomite, and magnesite. The data obtained indicate that alkali-bearing carbonate melts quench to the alkaline earth and double carbonates that are thermodynamically stable at quenching pressure and can be used as markers reflecting the pressure of their entrapment. Further, in this study, we established the fields of melt compositions corresponding to the distinct quench assemblages of carbonate minerals, which can be used for the reconstruction of the composition of carbonatitic melts entrapped by mantle minerals.

**Keywords:** carbonate melt; inclusions; kimberlite; diamond; bütschliite; Raman spectroscopy; multianvil experiment; Earth's mantle





## 1. Introduction

Inclusions of alkali-rich carbonatite melts are found in diamonds [1–3], magmatic minerals from lamproites [4], and minerals from mantle xenoliths carried by kimberlites [5–7]. A high concentration of potassium, up to 10–30 wt% $K_2O$, in carbonatite melt/fluid inclusions in diamonds was also detected [3,8–10]. However, the criteria to estimate the depth of entrapment and reconstruct the bulk melt composition need to be developed.

Alkaline carbonatite inclusions often contain double alkali–alkaline earth carbonates as daughter phases. Previously, a wide range of K-Ca-Mg double carbonates was found experimentally [11–16]. It was found that their stoichiometry and crystal structure change with increasing pressure [11,17,18]. For example, $K_2Ca_3(CO_3)_2$ at 3 GPa crystallizes in a $P2_12_12_1$ (ordered) [11], while at 6 GPa its structure transforms to a $Pnam$ space group (disordered) [19].

Thus, the following questions are raised: (1) Can K-Ca-Mg carbonates reflect the pressure at which inclusion was trapped? (2) Can quench assemblages reflect the bulk composition of the melt? To answer these questions, we studied the quench products of carbonate melts in the samples, synthesized previously in the $K_2CO_3$–$CaCO_3$–$MgCO_3$ system at 3 and 6 GPa [20,21] using microprobe analysis and Raman spectroscopy.

## 2. Materials and Methods

The experiments were carried out using a 'Discoverer-1500' DIA-type multianvil press at IGM SB RAS in Novosibirsk, Russia. "Fujilloy N-05" 26-mm tungsten carbide cubes with 12-mm truncations were employed as inner stage anvils. Pressure media were made of semisintered $ZrO_2$ ceramics (OZ-8C, MinoYogyo Co., Ltd., Gifu, Japan [22]) shaped as a 20.5-mm octahedron with ground edges and corners. Pyrophyllite gaskets, 4.0 mm in both width and thickness were used to seal the compressed volume and support the anvil flanks.

The heating was achieved using a tubular graphite heater, 4.0/4.5 mm-inner/outer diameter and 11 mm-length. The sample temperature was monitored via a W97Re3-W75Re25 thermocouple (C&T factory Co., Ltd., Tokyo, Japan) inserted in the heater center via walls and electrically insulated by $Al_2O_3$ tubes. No correction for the effect of pressure on the thermocouple electromotive force was applied.

High-temperature pressure calibration was performed using known phase transitions in $SiO_2$ (quartz–coesite) [23] and $CaGeO_3$ (garnet–perovskite) [24]. Uncertainty in the temperature and pressure measurements was estimated to be <25 °C and <0.5 GPa, respectively [25].

The experiments were performed by compression to a target press load corresponding to 3 or 6 GPa and then heating to a target temperature at a rate of 25–50 °C/min. The temperature was maintained within 2 °C of the desired value in a temperature control mode at a constant press load. The experiments were finished by turning off the heater power, resulting in a temperature drop to ambient in a few seconds, followed by slow decompression.

Starting materials were prepared by blending reagent grade $K_2CO_3$, $CaCO_3$, and natural magnesite (<0.1 mol% impurities) from Brumado (Bahia, Brazil) in an agate mortar with acetone and loaded as a powder into graphite cassettes. Since $K_2CO_3$ is a hygroscopic material, special attention to sample preparation was given to minimize the amount of moisture in the sample absorbed from the atmosphere. The loaded cassettes were dried at 300 °C for 1–2 h. Prepared assemblies were stored at 200 °C in a vacuum for $\geq$ 12 h prior to the experiment. All experiments were conducted at 15%–35% indoor humidity.

Composition and phase relations of the quench products were studied using a MIRA 3 LMU scanning electron microscope (Tescan Orsay Holding, Brno, Czech Republic) coupled with an INCA energy-dispersive X-ray microanalysis system 450 equipped with the liquid-nitrogen-free Large area EDS X-Max-80 Silicon Drift Detector (Oxford Instruments Nanoanalysis Ltd., Bognor Regis, UK) (SDD-EDS) at IGM SB RAS [26,27]. Energy-dispersive X-ray spectra were recorded by scanning a selected surface at an accelerating voltage of 20 kV and a current of 1 nA. The accumulation time of the spectra was 20 s.

The Raman spectra of the quench products of carbonate melts were recorded on a Horiba Jobin Yvon LabRAM HR800 spectrometer (HORIBA., Ltd., Kyoto, Japan) equipped with a multichannel LN/CCD detector with a resolution of 1024 pixels and a solid-state laser wavelength of 532.1 nm. The spectra were recorded at a beam power of 1–10 mW at the sample surface. The signal accumulation time was 100 s for the spectral range 50–1800 $cm^{-1}$. An Olympus BX41 microscope with a backscattering geometry was used as an optical system together with an Olympus Plan N100 $\times$ objective (working distance 0.2 mm, numerical aperture 0.8), which provided a focal spot diameter of 2 μm on the sample surface. The spectral resolution was ~1 $cm^{-1}$. This resolution was achieved through the use of an 1800-line/mm grating with 50-μm equivalent slits and a confocal aperture. The positions of the peaks in the obtained spectra were determined using the Gaussian function in the Fytik software [28].

## 3. Results

The backscattered electron images of quench phases are shown in Figure 1. The composition of the quench products and the bulk compositions of the analyzed melts are plotted on the *T-X* diagrams (Figure 2a–e) and are given in Tables 1 and 2. The

Raman spectra of the obtained phases are shown in Figure 3. The K2#/Ca# ratio, where K2# = $K_2CO_3/(CaCO_3 + MgCO_3 + K_2CO_3)$ and Ca# = $CaCO_3/(CaCO_3 + MgCO_3)$, is presented as numbers.

Run №; K$_2$#/Ca#, mol%; temperature; duration.

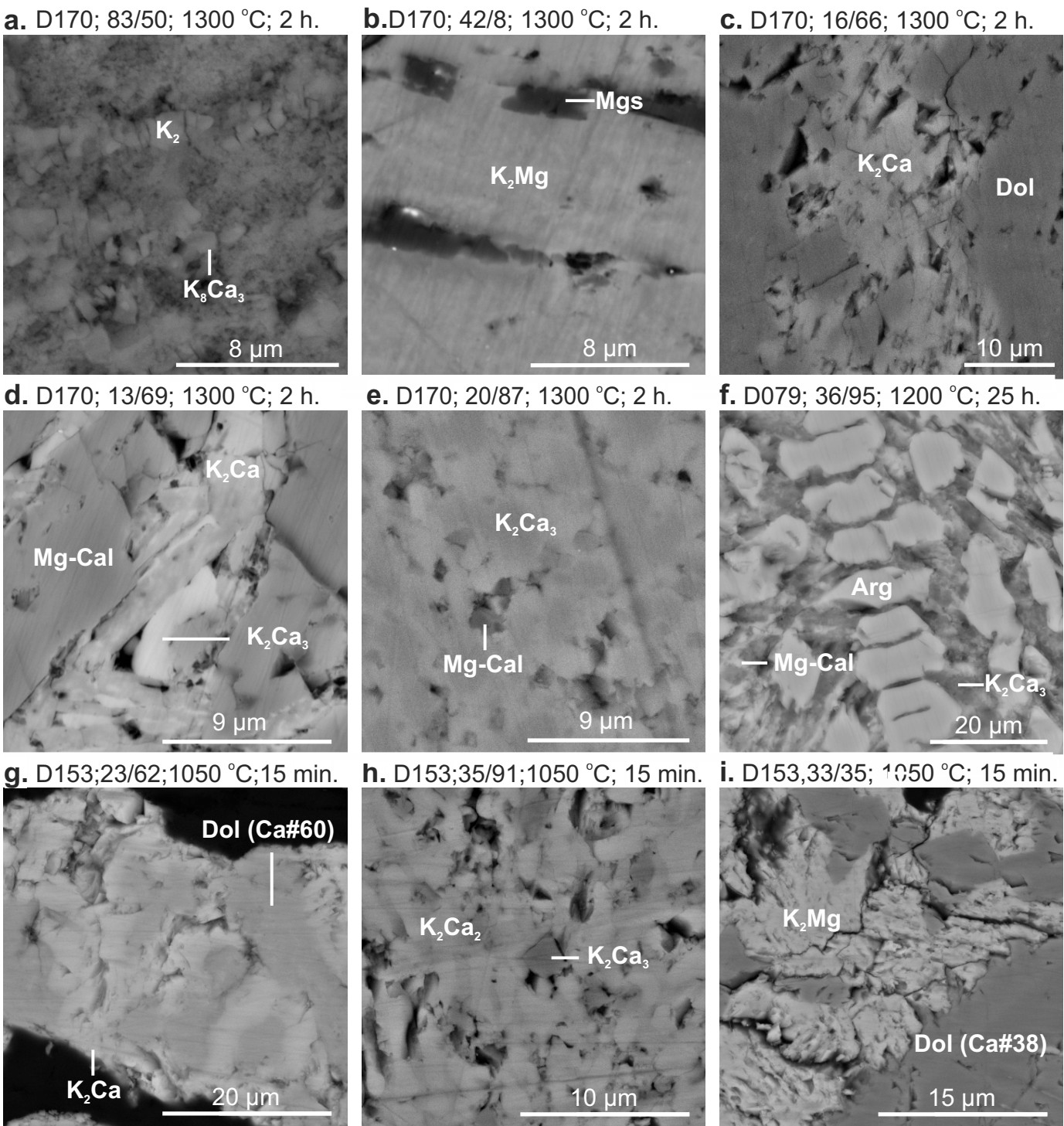

**Figure 1.** Backscattered electron images of quench products were obtained in the system $K_2CO_3$–$CaCO_3$–$MgCO_3$ at 6 (**a**–**f**) and 3 (**g**–**i**) GPa. $K_2$—solid solution of $CaCO_3$ in $K_2CO_3$; $K_2Ca$—$K_2Ca(CO_3)_2$ bütschliite; $K_2Ca_2$—$K_2Ca_2(CO_3)_3$; $K_2Ca_3$—$K_2Ca_3(CO_3)_4$; Arg—aragonite; Mg-Cal—Mg-bearing calcite; Dol—dolomite; Mgs—magnesite.

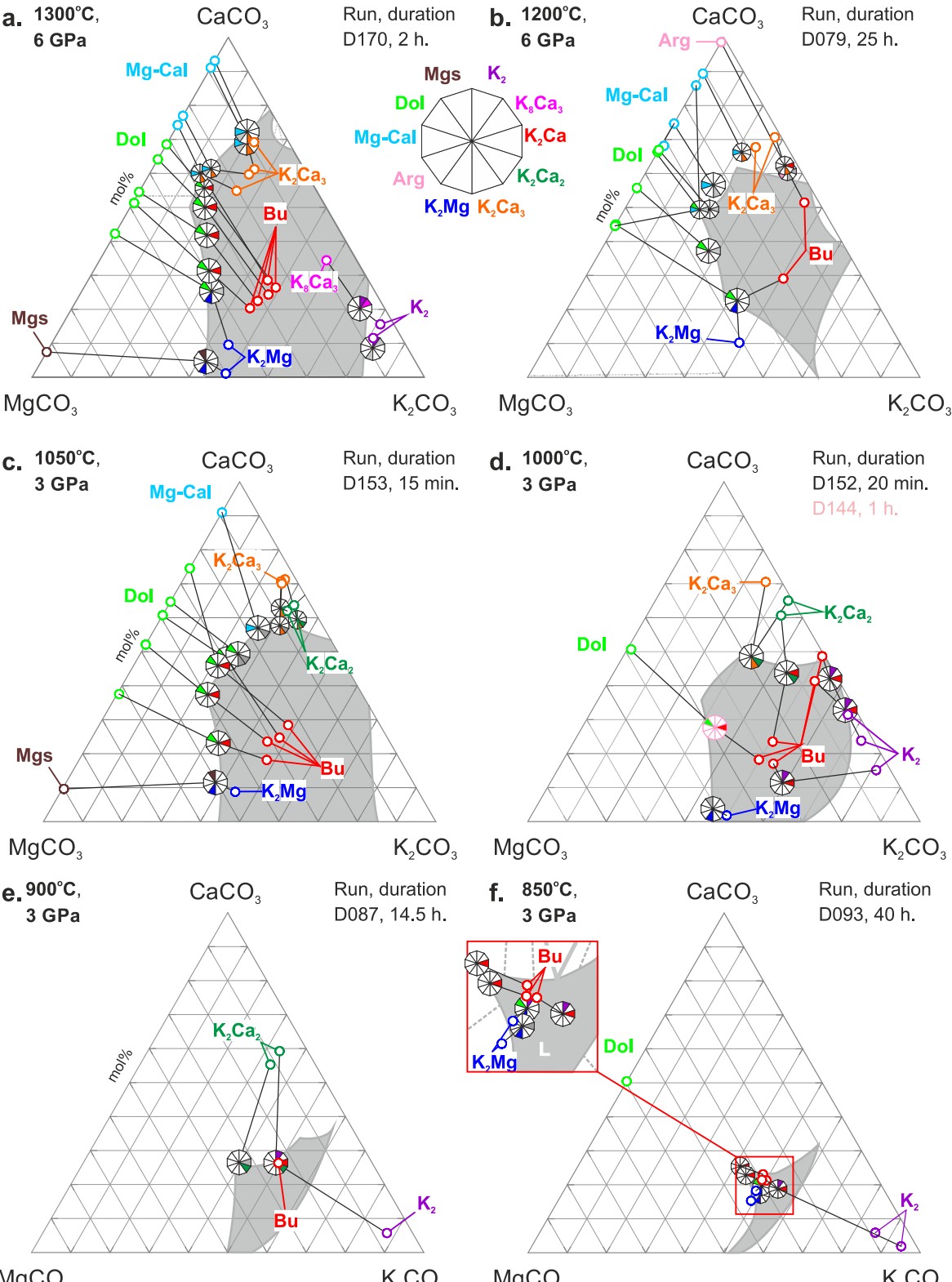

**Figure 2.** Compositions of quench phases of the melt from the $K_2CO_3$–$CaCO_3$–$MgCO_3$ at 6 (**a**,**b**) and 3 (**c**–**f**) GPa. $K_2$—solid solution of $CaCO_3$ in $K_2CO_3$; Bu—$K_2Ca(CO_3)_2$ bütschliite; $K_2Ca_2$—$K_2Ca_2(CO_3)_3$; $K_2Ca_3$—$K_2Ca_3(CO_3)_4$; Arg—aragonite; Mg-Cal—Mg-bearing calcite; Dol—dolomite; Mgs—magnesite. Gray polygons indicate the bulk composition of the melt. Colored segments indicate observed phases by SDD-EDS. Colored circles indicate the composition of quench phases, measured by EDS. The liquidus fields are shown in gray color.

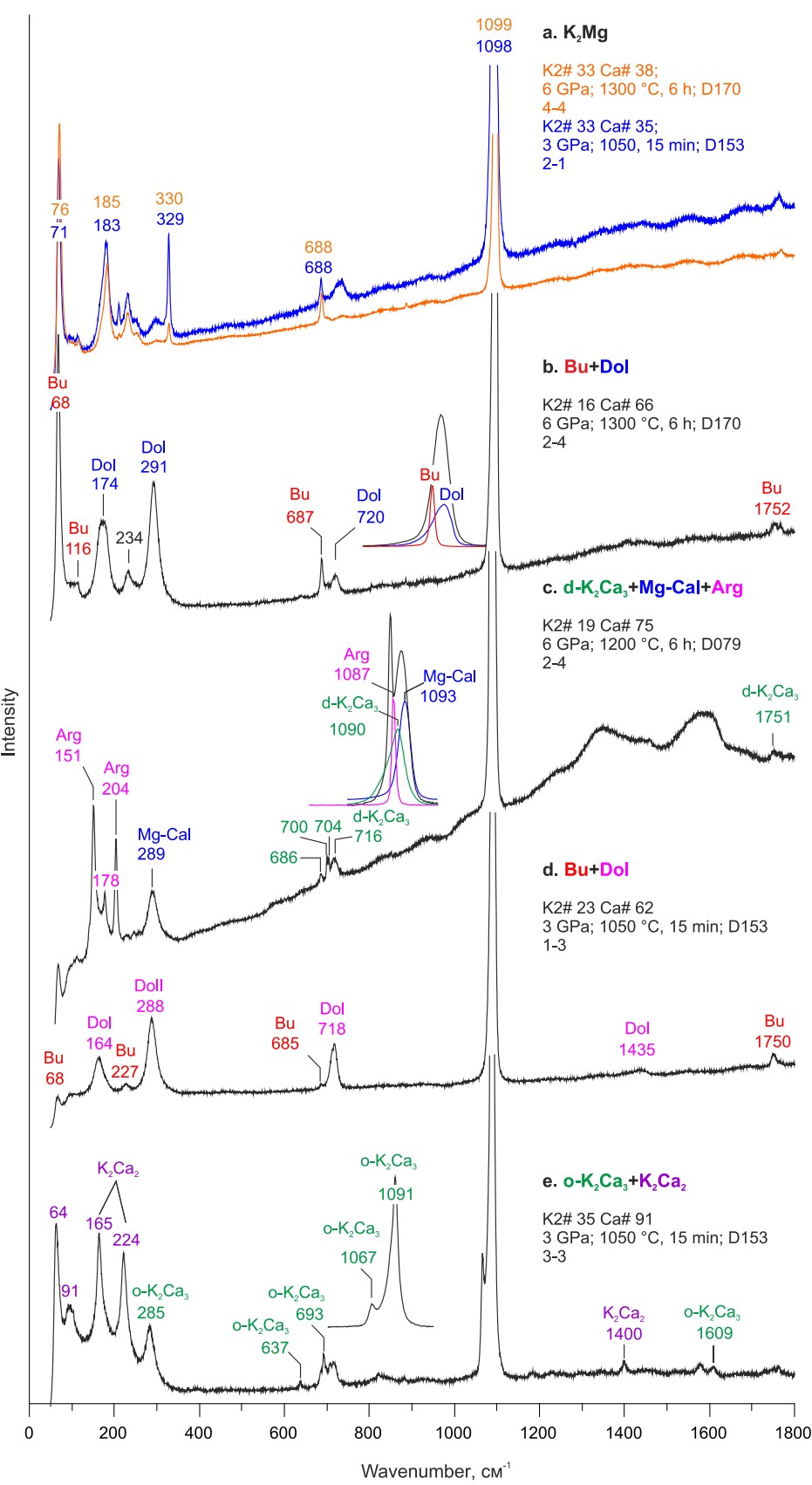

**Figure 3.** Raman spectra of quench products of the $K_2CO_3$-$CaCO_3$-$MgCO_3$ melts at 3 and 6 GPa. Raman spectra were collected at ambient conditions.



**Table 1.** Composition of quench products of $K_2CO_3$–$CaCO_3$–$MgCO_3$ melt at 6 GPa.

| Run | T, °C | t,h | # | Phase | *n* | $K_2CO_3$ | $\sigma$ | $CaCO_3$ | $\sigma$ | $MgCO_3$ | $\sigma$ | Ca# | $\sigma$ |
|---|---|---|---|---|---|---|---|---|---|---|---|---|---|
| D170 | 1300 | 2 | 1-1 | L | − | 82.6 | − | 8.6 | − | 8.8 | − | 49.6 | − |
| | | | | $K_2$ | 2 | 81.2 | 0.6 | 11.6 | 1.8 | 7.3 | 1.1 | 61.3 | 7.4 |
| | | | | $K_8Ca_3$ | + | + | + | + | + | + | + | + | + |
| | | | 1-3 | L | 4 | 42.1 | 0.9 | 4.6 | 4.6 | 53.3 | 0.9 | 8.0 | 0.5 |
| | | | | $K_2Mg$ | 8 | 48.8 | 0.6 | 1.1 | 0.3 | 50 | 0.6 | 2.2 | 0.6 |
| | | | | Mgs | + | + | + | + | + | + | + | + | + |
| | | | 2-1 | L | − | 20.2 | − | 69.0 | − | 10.8 | − | 86.5 | − |
| | | | | $K_2Ca_3$ | 21 | 26.1 | 0.9 | 61.2 | 0.9 | 12.7 | 0.8 | 82.8 | 1.0 |
| | | | | Mg-Cal | 8 | b.d.l. | − | 91.1 | 0.8 | 8.9 | 0.8 | 91.1 | 0.8 |
| | | | | Arg | + | + | + | + | + | + | + | + | + |
| | | | 2-2 | L | 2 | 18.8 | 1.3 | 72.3 | 0.3 | 9.0 | 0.0 | 88.9 | 0.0 |
| | | | | $K_2Ca_3$ | 2 | 22.0 | 0.5 | 69.2 | 1.5 | 8.8 | 1.0 | 88.7 | 1.4 |
| | | | | Mg-Cal | 4 | 8.0 | 1.3 | 85.7 | 1.5 | 6.3 | 0.2 | 93.2 | 0.3 |
| | | | | Arg | + | + | + | + | + | + | + | + | + |
| | | | 2-3 | L | 2 | 12.9 | 1.4 | 60.0 | 1.0 | 27.1 | 0.5 | 68.9 | 0.0 |
| | | | | Bu | 1 | 45.8 | − | 28.6 | − | 25.7 | − | 52.7 | − |
| | | | | $K_2Ca_3$ | 1 | 25.4 | − | 59.7 | − | 14.9 | − | 80.1 | − |
| | | | | Mg-Cal | 3 | b.d.l. | − | 74.2 | 1.3 | 25.8 | 1.3 | 74.2 | 1.3 |
| | | | 2-4 | L | 2 | 16.1 | 0.6 | 55.7 | 0.3 | 28.2 | 0.2 | 66.4 | 0.1 |
| | | | | Bu | 7 | 48.1 | 0.6 | 24.4 | 0.3 | 27.6 | 0.2 | 46.9 | 0.1 |
| | | | | Dol | 12 | b.d.l. | − | 68.6 | 2.0 | 31.4 | 2.0 | 68.6 | 2.0 |
| | | | 3-1 | L | − | 19.1 | − | 50 | − | 30.9 | − | 61.9 | − |
| | | | | Bu | 2 | 48.9 | 1.2 | 26.4 | 3.6 | 24.7 | 4.7 | 51.8 | 8.1 |
| | | | | Dol | 11 | b.d.l. | − | 64.1 | 2.0 | 35.9 | 2.0 | 64.1 | 2.0 |
| | | | 3-4 | L | 1 | 50.0 | − | 21.0 | − | 29.0 | − | 42.0 | − |
| | | | | Bu | 3 | 45.4 | 1.2 | 20.3 | 0.9 | 34.3 | 0.8 | 37.2 | 1.2 |
| | | | | Dol | 5 | 0.4 | 0.5 | 51.3 | 0.5 | 48.3 | 0.6 | 51.5 | 0.5 |
| | | | 3-3 | L | 2 | 15.0 | 0.0 | 61.4 | 0.3 | 23.6 | 0.3 | 72.2 | 0.3 |
| | | | | $K_2Ca_3$ | 8 | 24.7 | 2.4 | 54.9 | 6.0 | 20.4 | 4.8 | 72.8 | 6.7 |
| | | | | Mg-Cal | 13 | b.d.l. | − | 77.0 | 1.3 | 23.0 | 1.3 | 77.0 | 1.3 |
| | | | 4-1 | L | 7 | 73.3 | 0.6 | 20.4 | 0.5 | 6.3 | 0.3 | 76.3 | 0.8 |
| | | | | $K_2$ | 6 | 81.0 | 2.0 | 15.6 | 1.8 | 3.4 | 0.8 | 82.1 | 3.5 |
| | | | | $K_8Ca_3$ | 2 | 57.8 | 1.8 | 34.5 | 0.4 | 7.7 | 1.5 | 81.7 | 2.7 |
| | | | | $K_2Mg$ | + | + | + | + | + | + | + | + | + |
| | | | 4-3 | L | 2 | 23.7 | 0.1 | 41.8 | 0.2 | 34.5 | 0.1 | 54.8 | 0.1 |
| | | | | Bu | 15 | 46.3 | 1.0 | 22.5 | 1.4 | 31.2 | 1.3 | 41.9 | 2.4 |
| | | | | Dol | 16 | b.d.l. | − | 54.6 | 0.8 | 45.4 | 0.8 | 54.6 | 0.8 |
| | | | 4-4 | L | 1 | 33.2 | − | 25.3 | − | 41.5 | − | 37.9 | − |
| | | | | $K_2Mg$ | 18 | 45.3 | 1.7 | 9.6 | 1.9 | 45.1 | 2.1 | 17.5 | 3.5 |
| | | | | Dol | 5 | b.d.l. | − | 42.3 | 2.4 | 57.7 | 2.4 | 42.3 | 2.4 |
| D079 | 1200 | 25 | 1-1 | L | − | 27.5 | − | 37.2 | − | 35.3 | − | 51.3 | − |
| | | | | Bu | + | + | + | + | + | + | + | + | + |
| | | | | Dol | 7 | b.d.l. | − | 57.7 | 0.9 | 42.3 | 0.9 | 57.7 | 0.9 |
| | | | 1-3 | L | 2 | 19.1 | 0.8 | 49.4 | 0.7 | 31.5 | 0.1 | 61.0 | 0.2 |
| | | | | Bu | + | + | + | + | + | + | + | + | + |
| | | | | Dol | 6 | b.d.l. | − | 44.4 | 1.7 | 55.6 | 1.7 | 44.4 | 1.7 |
| | | | | Dol(2) | 22 | b.d.l. | − | 64.3 | 1.6 | 35.7 | 1.6 | 64.3 | 1.6 |
| | | | | Mg-Cal | 3 | b.d.l. | − | 85.6 | 3.8 | 14.4 | 3.8 | 85.6 | 3.8 |
| | | | 1-4 | L | 2 | 21.7 | 0.6 | 49.5 | 1.3 | 28.8 | 0.7 | 63.2 | 1.2 |
| | | | | Bu | + | + | + | + | + | + | + | + | + |
| | | | | Dol | 11 | b.d.l. | − | 68.2 | 0.9 | 31.9 | 0.9 | 68.6 | 0.9 |
| | | | 2-2 | L | 1 | 42.2 | − | 22.6 | − | 35.3 | − | 39.0 | − |
| | | | | $K_2Mg$ | 4 | 48.9 | 0.3 | 10.2 | 0.7 | 40.9 | 0.7 | 20.0 | 1.4 |
| | | | | Bu | 2 | 50.7 | 0.4 | 29.1 | 1.8 | 20.2 | 2.2 | 59.0 | 4.2 |
| | | | | Dol | 1 | b.d.l. | − | 45 | − | 55 | − | 45 | − |

**Table 1.** *Cont.*

| Run | T, °C | t,h | # | Phase | n | K₂CO₃ | σ | CaCO₃ | σ | MgCO₃ | σ | Ca# | σ |
|---|---|---|---|---|---|---|---|---|---|---|---|---|---|
| | | | 2-3 | L | 3 | 21.5 | 1.2 | 66.3 | 1.0 | 12.3 | 0.3 | 84.4 | 0.2 |
| | | | | K₂Ca₃ | 19 | 24.3 | 0.5 | 67.8 | 3.4 | 8.0 | 3.7 | 89.5 | 4.8 |
| | | | | Mg-Cal | 2 | b.d.l. | — | 89.4 | 5.2 | 10.6 | 5.2 | 89.4 | 5.2 |
| | | | 3-2 | L | — | 35.9 | — | 60.6 | — | 3.4 | — | 94.6 | — |
| | | | | K₂Ca₃ | + | + | + | + | + | + | + | + | + |
| | | | | Arg | 6 | b.d.l. | - | 98.7 | 0.5 | 1.3 | 0.5 | 98.7 | 0.5 |
| | | | | Mg-Cal | + | + | + | + | + | + | + | + | + |
| | | | 3-3 | L | 1 | 34.4 | | 62.3 | | 3.3 | | 96.0 | |
| | | | | Bu | 2 | 44.8 | 0.4 | 51.5 | 0.8 | 3.7 | 1.2 | 93.3 | 2.2 |
| | | | | K₂Ca₃ | — | 27.7 | — | 70.7 | — | 1.6 | — | 97.8 | — |
| | | | | Arg | + | + | + | + | + | + | + | + | + |

Notes: *n*—numbers of analysis; *σ*—standard deviation; #—sample number; L—the bulk composition of the melt; K₂—solid solution of CaCO₃ in K₂CO₃; Bu—K₂Ca(CO₃)₂ bütschliite; K₂Ca₂—K₂Ca₂(CO₃)₃; K₂Ca₃—K₂Ca₃(CO₃)₄; Arg—aragonite; Dol—dolomite; Mg-Cal—Mg-bearing calcite; Mgs—magnesite; Ca# = 100·Ca₂CO₃/(Mg₂CO₃ + CaCO₃); "+"—phase determined by Raman spectroscopy, but its composition is undetermined by EDS; "−"—no data.

**Table 2.** Composition of quench products of K₂CO₃–CaCO₃–MgCO₃ melt at 3 GPa.

| Run | T, °C | t,h | # | Phase | n | K₂CO₃ | σ | CaCO₃ | σ | MgCO₃ | σ | Ca# | σ |
|---|---|---|---|---|---|---|---|---|---|---|---|---|---|
| D153 | 1050 | 0.25 | 1-3 | L | 1 | 22.7 | — | 48.1 | — | 29.2 | — | 62.2 | — |
| | | | | Bu | 2 | 48.2 | 0.4 | 28.4 | 1.2 | 23.4 | 1.6 | 54.8 | 2.7 |
| | | | | Dol | 2 | b.d.l. | — | 74.6 | 0.3 | 25.4 | 0.3 | 74.6 | 0.3 |
| | | | 2-1 | L | 1 | 32.8 | — | 23.4 | — | 43.8 | — | 34.8 | — |
| | | | | Bu | 12 | 47.8 | 0.6 | 18.1 | 1.4 | 34.0 | 1.2 | 34.8 | 2.5 |
| | | | | Dol | 8 | 0.5 | 0.0 | 37.5 | 1.8 | 63.0 | 1.8 | 37.7 | 1.8 |
| | | | 2-2 | L | 1 | 38.2 | — | 11.4 | — | 50.4 | — | 18.5 | — |
| | | | | K₂Mg | 11 | 44.5 | 1.2 | 8.8 | 0.7 | 46.8 | 0.8 | 15.8 | 1.0 |
| | | | | Mgs | 1 | b.d.l. | — | 10 | — | 90 | — | 10 | — |
| | | | 3-2 | L | 1 | 29.7 | — | 60.9 | — | 9.4 | — | 86.7 | — |
| | | | | K₂Ca₂ | + | + | + | + | + | + | + | + | + |
| | | | | K₂Ca₃ | 13 | 25.8 | 0.7 | 69.9 | 1.3 | 4.4 | 0.8 | 94.1 | 1.1 |
| | | | 3-3 | L | 1 | 35.0 | — | 59.4 | — | 5.6 | — | 91.4 | — |
| | | | | K₂Ca₂ | 1 | 32.0 | — | 63.7 | — | 4.3 | — | 93.7 | — |
| | | | | K₂Ca₃ | 1 | 25.9 | — | 71.3 | — | 2.8 | — | 96.2 | — |
| | | | 3-4 | L | 1 | 26.2 | — | 57.0 | — | 16.9 | — | 77.1 | — |
| | | | | K₂Ca₂ | + | + | + | + | + | + | + | + | + |
| | | | | Mg-Cal | 1 | b.d.l. | — | 91.0 | — | 9.0 | — | 91.0 | — |
| | | | 4-1 | L | 1 | 30.0 | — | 59.7 | — | 10.5 | — | 85.0 | — |
| | | | | K₂Ca₂ | 2 | 31.1 | 0.0 | 62.0 | 0.6 | 6.9 | 0.5 | 90.0 | 0.8 |
| | | | | K₂Ca₃ | 12 | 25.2 | 0.9 | 70.7 | 3.4 | 4.1 | 2.6 | 94.5 | 3.6 |
| | | | 4-2 | L | 1 | 24.8 | — | 49.5 | — | 25.6 | — | 65.9 | — |
| | | | | K₂Ca₂ | + | + | + | + | + | + | + | + | + |
| | | | | K₂Ca₃ | + | + | + | + | + | + | + | + | + |
| | | | | Dol | 1 | b.d.l. | — | 64.7 | — | 35.3 | — | 64.7 | — |
| | | | 4-3 | L | 1 | 24.8 | — | 49.5 | — | 25.6 | — | 65.9 | — |
| | | | | Bu | 1 | 47.8 | — | 24.7 | — | 27.5 | — | 47.4 | — |
| | | | | Dol | 1 | b.d.l. | — | 64.7 | — | 35.3 | — | 64.7 | — |
| | | | 4-4 | L | 1 | 23.1 | — | 37.4 | — | 39.6 | — | 48.6 | — |
| | | | | Bu | 1 | 47.8 | — | 24.7 | — | 27.5 | — | 47.4 | — |
| | | | | Dol | 3 | b.d.l. | — | 52.1 | 0.4 | 47.9 | 0.4 | 52.1 | 0.4 |
| D152 | 1000 | 0.3 | 1-3 | L | 1 | 45.4 | — | 3.8 | — | 48.0 | — | 7.0 | — |
| | | | | K₂Mg | 6 | 47.9 | 0.1 | 4.0 | 0.4 | 48.0 | 0.5 | 7.7 | 0.8 |
| | | | | Mgs | + | + | + | + | + | + | + | + | + |
| | | | 2-3 | L | 1 | 59.2 | — | 11.4 | — | 29.4 | — | 28.0 | — |
| | | | | K₂ | 1 | 93.2 | — | 5.6 | — | 1.2 | — | 82.9 | — |
| | | | | Bu | 1 | 53.0 | — | 17.5 | — | 29.6 | — | 37.1 | — |

**Table 2.** *Cont.*

| Run | T, °C | t,h | # | Phase | n | K$_2$CO$_3$ | σ | CaCO$_3$ | σ | MgCO$_3$ | σ | Ca# | σ |
|---|---|---|---|---|---|---|---|---|---|---|---|---|---|
| | | | 2-4 | L | 1 | 64.5 | − | 32.7 | − | 2.8 | − | 92.1 | − |
| | | | | K$_2$ | 3 | 68.3 | 1.7 | 29.0 | 2.3 | 2.8 | 1.2 | 91.2 | 2.5 |
| | | | | Bu | 3 | 53.2 | 0.3 | 44.6 | 0.5 | 2.2 | 0.4 | 95.2 | 0.8 |
| | | | 3-1 | L | 1 | 55.8 | − | 42.2 | − | 2.0 | − | 95.4 | − |
| | | | | K$_2$ | 3 | 73.1 | 1.5 | 23.8 | 1.1 | 3.1 | 0.3 | 88.4 | 0.7 |
| | | | | Bu | 3 | 50.6 | 0.2 | 48.8 | 0.2 | 0.6 | 0.1 | 98.7 | 0.1 |
| | | | 3-3 | L | 1 | 44.1 | − | 43.7 | − | 12.2 | − | 78.3 | − |
| | | | | Bu | 4 | 49.5 | 1.8 | 26.0 | 1.7 | 24.5 | 2.1 | 51.5 | 3.3 |
| | | | | K$_2$Ca$_2$ | 4 | 34.2 | 0.5 | 60.7 | 2.2 | 5.1 | 1.9 | 92.2 | 2.9 |
| | | | 3-4 | L | 1 | 32.7 | − | 48.5 | − | 18.8 | − | 72.1 | − |
| | | | | K$_2$Ca$_2$ | 1 | 33.3 | − | 65.5 | − | 1.2 | − | 98.2 | − |
| | | | | K$_2$Ca$_3$ | 1 | 25.4 | − | 70.5 | − | 4.1 | − | 94.5 | − |
| D144 | 1000 | 1 | 1-4 | L | 1 | 33.9 | − | 27.7 | − | 38.5 | − | 41.8 | − |
| | | | | Bu | 1 | 50.0 | − | 18.2 | − | 31.8 | − | 36.4 | − |
| | | | | Dol | 7 | 0.8 | 0.1 | 50.7 | 1.3 | 48.5 | 1.3 | 51.1 | 1.3 |
| D087 | 900 | 14.5 | 2-3 | L | 2 | 39.3 | 0.0 | 27.1 | 0.1 | 33.5 | 0.1 | 44.7 | 0.1 |
| | | | | K$_2$Ca$_2$ | 1 | 33.2 | − | 55.3 | − | 11.5 | − | 82.8 | − |
| | | | | Bu | + | + | + | + | + | + | + | + | + |
| | | | 3-2 | L | 1 | 49.1 | − | 26.4 | − | 24.5 | − | 51.8 | − |
| | | | | K$_2$ | 2 | 87.7 | 1.1 | 5.7 | 0.6 | 6.6 | 0.6 | 46.6 | 0.3 |
| | | | | Bu | 5 | 49.6 | 1.3 | 25.4 | 1.6 | 25.0 | 1.7 | 50.4 | 2.9 |
| | | | | K$_2$Ca$_2$ | 1 | 33.6 | − | 59.2 | − | 7.1 | − | 89.2 | − |
| D093 | 850 | 40 | 1-3 | L | 1 | 50.6 | − | 17.2 | − | 32.2 | − | 34.9 | − |
| | | | | K$_2$Mg | 3 | 49.1 | 0.4 | 18.2 | 0.6 | 32.8 | 0.6 | 35.7 | 1.0 |
| | | | | Dol | 1 | b.d.l. | − | 50.4 | − | 49.6 | − | 50.4 | − |
| | | | | K$_2$ | 3 | 85.7 | 2.7 | 5.8 | 1.3 | 8.5 | 1.4 | 40.6 | 2.1 |
| | | | 1-4 | L | 1 | 54.4 | − | 18.7 | − | 26.9 | − | 41.0 | − |
| | | | | K$_2$ | 1 | 94.3 | − | 1.8 | − | 3.9 | − | 32.0 | − |
| | | | | Bu | 6 | 50.2 | 0.7 | 21.2 | 2.0 | 28.6 | 2.4 | 42.6 | − |
| | | | 3-1 | L | 1 | 42.9 | − | 24.4 | − | 32.7 | − | 42.8 | − |
| | | | | Bu | 1 | 48.4 | − | 23.0 | − | 28.5 | − | 44.7 | − |
| | | | 3-4 | L | 1 | 41.9 | − | 25.6 | − | 32.5 | − | 44.1 | − |
| | | | | K$_2$ | + | + | + | + | + | + | + | + | + |
| | | | | K$_2$Mg | 1 | 49.3 | − | 15.2 | − | 35.5 | − | 30.1 | − |
| | | | 4-4 | L | 1 | 42.9 | − | 23.6 | − | 33.5 | − | 41.3 | − |
| | | | | Bu | 3 | 48.4 | 1.4 | 22.8 | 0.6 | 28.8 | 1.3 | 44.2 | 1.2 |

Notes: see Table 1.

### 3.1. 6 GPa

At 1300 °C, twelve melts with bulk ratios K2#/Ca#: 83/50, 73/76, 42/8, 33/38, 50/42, 24/55, 19/62, 16/66, 13/69, 20/87, and 19/89 were studied (Figure 2a). At 83/50 and 73/76, the quench products are represented by isometric grains of K$_8$Ca$_3$(CO$_3$)$_7$ (2 × 2 μm) and elongated crystals of K$_2$CO$_3$ (8 × 3 μm) embedded in a fine dendritic aggregate (Figure 1a, Table 1). Dendritic crystals of K$_2$Mg(CO$_3$)$_2$ contain small inclusions of magnesite (4 × 4 μm) in the melt with 42/8 (Figure 1b, Table 1) or dolomite needles in the melt with 33/38 (Table 1). The Raman spectrum of quenched K$_2$Mg(CO$_3$)$_2$ contains bands at 76, 185, 330, 688, and 1099 cm$^{-1}$ corresponding to subsolidus K$_2$Mg(CO$_3$)$_2$-$R\bar{3}m$, which was synthesized at 3 and 6 GPa [18,29] (Figure 3a (orange)). At 50/42, 24/55, 19/62, and 16/66, the quench products consist of K$_2$Ca(CO$_3$)$_2$ and dolomite (Figure 1c, Table 1). The Raman spectrum of the quenched melt with 16/66 has bands at 68, 116, 687, and 1752 cm$^{-1}$, corresponding to bütschliite-$R\bar{3}2/m$ [30], synthesized at 1 atm, and 3 and 6 GPa [11,31], and bands at 174, 291, 720 cm$^{-1}$, which are characteristic to dolomite [32] (Figure 3b). The quench products of melt with 13/69 are represented by the K$_2$Ca(CO$_3$)$_2$ + K$_2$Ca$_3$(CO$_3$)$_4$ + Mg-calcite assemblage (Figure 1d, Table 1). The K$_2$Ca$_3$(CO$_3$)$_4$ + Mg-Cal assemblage was found in the melt with 20/87 and 19/89 (Figure 1e, Table 1).

Quench products of nine melts with bulk ratios K2#/Ca#: 28/51, 19/61, 22/63, 42/39, 22/84, 19/75, 36/95, and 35/96 were studied at 1200 °C (Figure 2b). The melt with 28/51 quenches to bütschliite and dolomite (Table 1). At 19/61, the quench products consist of Dol (Ca#44), Dol (Ca#64), $K_2Ca(CO_3)_2$, and Mg-Cal (Table 1). Bütschliite and dolomite are found in the melt with 22/63 (Table 1). At 42/39, the quench products are represented by $K_2Mg(CO_3)_2 + K_2Ca(CO_3)_2$ + Dol assemblage (Table 1). The quench products of melt with 22/84 and 19/75 consist of $K_2Ca_3(CO_3)_4$ and Mg-calcite (Table 1). The Raman spectrum of the melt with 19/75 contains bands at 686, 700, 704, 716, 1090, 1751 cm$^{-1}$, which correspond to d-$K_2Ca_3(CO_3)_4$ with the Pnam structure [11]; bands at 289 and 1093 cm$^{-1}$, which match to Mg-bearing calcite [32]; and bands at 151, 178, 204, 1087 cm$^{-1}$, which coincide with that of aragonite [33] (Figure 3c). Coarse grains of Arg (15 × 15 μm) surrounded by the fine aggregate of Mg-calcite and $K_2Ca_3(CO_3)_4$ were found in the melt with 36/95 (Figure 1f, Table 1). The quench products of melt with 35/96 consist of $K_2Ca(CO_3)_2$, $K_2Ca_3(CO_3)_4$, and aragonite (Table 1).

*3.2. 3 GPa*

At 3 GPa, the quench products of carbonate melts were studied in the temperature range of 850–1050 °C (Figure 2c–e). At 1050 °C, quench products were studied in ten melts with bulk ratios K2#/Ca#: 23/48, 23/62, 25/66, 33/35, 30/87, 29/88, 32/94, 31/90, 38/19, and 26/77 (Figure 2c). At 23/48, 23/62, 25/66, and 33/35, quench products are represented by $K_2Ca(CO_3)_2$ + Dol assemblage (Figure 1g, Table 2). The Raman spectrum of the melt with 23/62 contains bands at 68, 227, 685, 1750 cm$^{-1}$ corresponding to bütschliite [11,31] and bands at 164, 288, 718, 1435 cm$^{-1}$ dolomite [32] (Figure 3b). Small equigranular grains of $K_2Ca_2(CO_3)_3$ (5 × 5 μm) and $K_2Ca_3(CO_3)_4$ (3 × 3 μm) were found among the quench products of melt with 30/87, 29/88, 32/94, and 31/90, respectively (Figure 1h, Table 2). The Raman spectrum of the melt at 35/91 contains bands at 64, 91, 165, 224, and 1400 cm$^{-1}$, which correspond to $K_2Ca_2(CO_3)_3$ (R3) [34], as well as bands at 285, 637, 693, 1067, 1091, and 1609 cm$^{-1}$ characteristic of o-$K_2Ca_3(CO_3)_4$ with an ordered structure (P2$_1$2$_1$2$_1$) [19], which were synthesized at 3 GPa [11] (Figure 3e). At 38/19, the quench products are represented by the $K_2Mg(CO_3)_4$ + magnesite assemblage (Table 2). The Raman spectrum of this melt contains bands that match to $K_2Mg(CO_3)_2$, which was previously synthesized at 3 and 6 GPa [18,29] (Figure 3a (dark blue)). At 26/77, the quench products consist of $K_2Ca_2(CO_3)_3$ and Mg-calcite (Table 2).

At 1000 °C, the quench products were studied in six melts with the following bulk ratios K2#/Ca#: 56/95, 59/28, 65/92, 44/78, 33/72, and 34/42 (Figure 2d). At 45/7, the melt quenches to $K_2Mg(CO_3)_2$ and magnesite (Table 2). At 56/95, 59/28, and 65/92, the quench products are represented by the $K_2Ca(CO_3)_2 + K_2CO_3$ assemblage (Table 2). $K_2Ca(CO_3)_2$ and $K_2Ca_2(CO_3)_3$ were found in the melt with 44/78 (Table 2). At 33/72, the quench products consist of $K_2Ca_2(CO_3)_3$ and $K_2Ca_3(CO_3)_4$ (Table 2). At 34/42, the melt quenches to bütschliite and dolomite (Table 2).

At 900 °C, the quench products of two melts with bulk ratios K2#/Ca#: 40/45 and 49/52 were studied (Figure 2e). At 40/45, $K_2Ca_2(CO_3)_3$ and $K_2Ca(CO_3)_2$ were found in the quench products (Table 2). At 49/52, the quench products are represented by the $K_2CO_3 + K_2Ca(CO_3)_2 + K_2Ca_2(CO_3)_3$ assemblage (Table 2).

Quench products of five melts with bulk ratios K2#/Ca#: 51/35, 54/41, 43/43, 43/41, and 42/44 were studied at 850 °C (Figure 2f). At 51/35, the quench products consist of $K_2Mg(CO_3)_2$ and dolomite (Table 2). At 54/41, the melt quenches to the $K_2CO_3 + K_2Ca(CO_3)_2$ (Table 2). At 43/43 and 43/41, the quench products are represented by $K_2Ca(CO_3)_2$, while at 42/44, by $K_2CO_3$ and $K_2Mg(CO_3)_2$ (Table 2).

## 4. Discussion

*4.1. K-Ca Carbonates as Pressure Markers*

Since carbonate melts quench to subsolidus phases that are thermodynamically stable at quenching pressures, the following carbonates can be considered as pressure markers.

$K_2Ca_3(CO_3)_4$ (disordered) appears at 6 GPa and K2# $\leq$ 40, Ca# $\geq$ 68 in the melt (Figure 2a,b). The 6-Gpa polymorph of $K_2Ca_3(CO_3)_4$ is orthorhombic with space group *Pnam*, and *a* = 7.53915(18) Å, *b* = 8.7799(2) Å, *c* = 16.1811(4) Å [11]. The Raman spectrum of disordered $K_2Ca_3(CO_3)_4$ is given in Figure 3a.

$K_8Ca_3(CO_3)_7$ crystallizes at 6 GPa and K2# $\geq$ 50, Ca# $\geq$ 50 in the melt (Figure 2a,b). $K_8Ca_3(CO_3)_7$ is trigonal (*P3m*1) with *a* = 16.3143(10) Å and *c* = 7.1146(4) Å [11]. The Raman spectrum of $K_8Ca_3(CO_3)_7$ is given in [11].

$K_2Ca_3(CO_3)_4$ (ordered) forms at 3 GPa and K2# $\leq$ 30, Ca# $\geq$ 70 in the melt (Figure 2c,d). The 3-Gpa polymorph of $K_2Ca_3(CO_3)_4$ is orthorhombic with space group $P2_12_12_1$, and *a* = 7.3905(2) Å, *b* = 8.8156(3) Å, *c* = 16.4817(5) Å [19]. The Raman spectrum of disordered $K_2Ca_3(CO_3)_4$ is given in [11].

$K_2Ca(CO_3)_2$ bütschliite was detected in the quench products at both 3 and 6 GPa. Unlike that, the appearance of $K_2Ca(CO_3)_2$ fairchildite can be expected at crustal pressures since it was established as a liquidus phase at 0.1 GPa [12–14]. Bütschliite crystallizes in the space group $R\bar{3}2/m$ (*a* = 5.38 Å, *c* = 18.12 Å, Z = 3) with an eitelite-type structure. Fairchildite crystallizes in the space group $P6_3/mmc$ (*a* = 5.294(1) Å, *c* = 13.355(2) Å). The Raman spectra of bütschliite and fairchildite are given in [11].

The listed carbonates have narrow fields of stability and their finds in mineral inclusions may reflect the lower or upper limits of the inclusion uptake pressure.

### 4.2. Implication for Carbonatite Inclusions

Logvinova et al. (2019) [1] found a carbonate inclusion in a gem-quality diamond. The Raman and EDS spectra from the daughter phases in this inclusion revealed the presence of bütschliite, dolomite, and eithelite. Since neither bütschliite nor eithelite can be in equilibrium with dolomite [20,35], the authors concluded that these carbonates were melted at the moment of capture. Moreover, the authors reported that the reconstructed bulk composition of this alkali–carbonatite melt inclusion corresponds to K2# 10 and Ca# 57. According to our experimental data on the system $K_2CO_3$–$CaCO_3$–$MgCO_3$ at 6 GPa, a melt of this composition is quenched into bütschliite and dolomite (Figure 4a), which is in good agreement with the observations of [1].

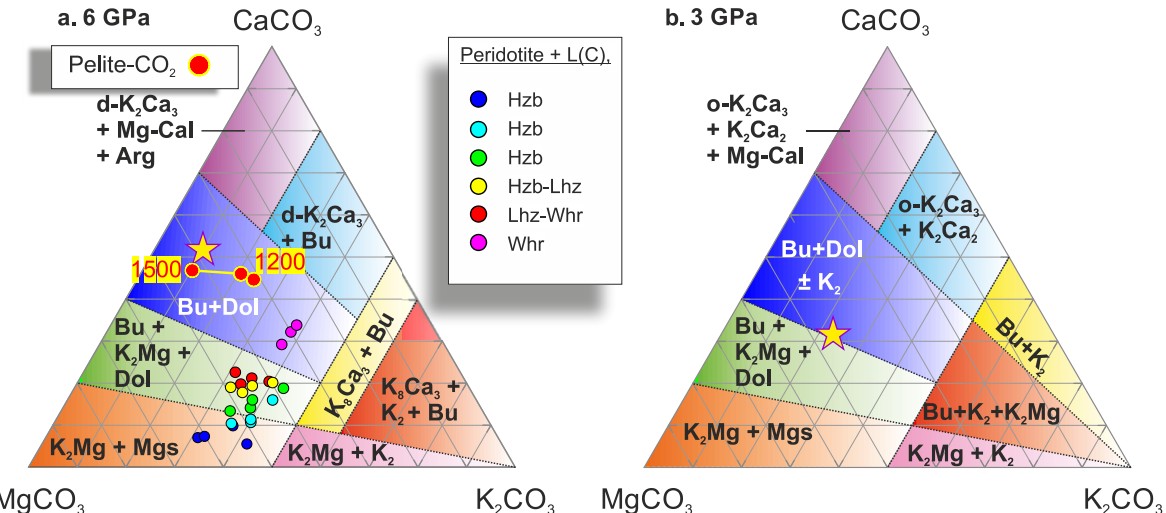

**Figure 4.** Assemblages of quench products at 6 (**a**) and 3 (**b**) GPa. Stars—carbonatite melt inclusion was reported by [1] (**a**) and [5] (**b**). Carbonatite melts from the complex systems: pelite-$CO_2$ [36], lherzolite + L(C) with Ca# 10–40 [37].

Giuliani et al. (2012) [5] found K-Ca carbonate (presumably $K_2Ca(CO_3)_2$ fairchildite) and dolomite among the daughter phases in a melt inclusion (DU1-2D-in1) in metasomatic ilmenite from ilmenite-rich domains emplaced in a spinel harzburgite wall rock. The *P-T*

conditions for the formation of this xenolith of spinel harzburgite are estimated to be 3.4–3.5 GPa and 860 °C. We estimated the approximate bulk composition of the inclusion based on the volume ratio of daughter phases reported by the authors. The obtained composition corresponds to K2# 23 and Ca# 41. According to the present results in the $K_2CO_3$–$CaCO_3$–$MgCO_3$ ternary at 3 GPa, the carbonate melt with such K2# and Ca# ratios quenches to bütschliite and dolomite (Figure 4b), which is consistent with the observations of [5] and suggests that K-Ca carbonate must be bütschliite rather than fairchildite.

Comparison of our experimental data with natural observations given above allows us to establish the fields of melt compositions corresponding to the distinct quench assemblages of carbonate minerals, which can be used for the reconstruction of the composition of carbonatitic melts entrapped by mantle minerals.

*4.3. Comparison of Quench Assemblages with Subsolidus Phase Fields of the $K_2CO_3$–$CaCO_3$–$MgCO_3$ System at 3 and 6 GPa*

According to the obtained results, the phase composition of the quench products depends on the K2# and Ca# ratios of the carbonate melt (Figure 4). In this regard, it seems interesting to compare quench and subsolidus assemblages in the $K_2CO_3$–$CaCO_3$–$MgCO_3$ system at 3 and 6 GPa [20,21]. The following subsolidus assemblages match the established quench products of K-Ca-Mg carbonate melt: magnesite + $K_2Mg(CO_3)_2$, $K_2Mg(CO_3)_2$ + $K_2CO_3$, aragonite + d-$K_2Ca_3(CO_3)_4$ + Ca-dolomite, and $K_8Ca_3(CO_3)_7$ + $K_2CO_3$ at 6 GPa (Figure 4a); magnesite + $K_2Mg(CO_3)_2$, $K_2Mg(CO_3)_2$ + $K_2CO_3$, $K_2Ca_2(CO_3)_3$ + $K_2Ca(CO_3)_2$, and $K_2Ca(CO_3)_2$ + $K_2CO_3$ at 3 GPa (Figure 4b). At the same time, there are metastable assemblages formed during melt quenching that were not found under equilibrium conditions below subsolidus, namely, $K_2Ca(CO_3)_2$ + dolomite and $K_2Ca(CO_3)_2$ + $K_2Mg(CO_3)_2$ + dolomite at 6 GPa and $K_2Ca(CO_3)_2$ + $K_2Mg(CO_3)_2$, $K_2Ca(CO_3)_2$ + dolomite, o-$K_2Ca_3(CO_3)_4$ + $K_2Ca_2(CO_3)_3$ + Mg-calcite at 3 GPa (Figure 4b). Thus, the K-Ca-Mg carbonate melt at 3 and 6 GPa quench to carbonates that are thermodynamically stable at subsolidus conditions, but their quench assemblages may not coincide to subsolidus phase fields. Further, assemblages of carbonates in carbonate inclusions corresponding to the quench phases established in this study may indicate the melting origin of these inclusions.

*4.4. Expected Quench Products of Mantle Melts from Complex Systems*

The compositions of carbonate melt generated by partial melting of carbonated pelite and peridotite at 6 GPa were plotted on the *T-X* phase diagram with fields of quench phases (Figure 4a) [36–38]. Ca# of melts formed in the pelite-$CO_2$ system at 6 GPa and 1200–1500 °C vary from 50 to 60; consequently, according to the *T-X* phase diagram, its quench products will contain bütschliite and dolomite. Bütschliite and dolomite are also expected among the quench products of carbonate melt, which are in equilibrium with the wehrlite with Ca# > 34 (Figure 4a). Carbonate melts in equilibrium with lherzolite have Ca# 30–34 and, therefore, should be quenched to the $K_2Ca(CO_3)_2$ + $K_2Mg(CO_3)_2$ + dolomite assemblage. At the same time, melts in equilibrium with harzburgite have Ca# 20–34 and have to quench to the bütschliite + $K_2Mg(CO_3)_2$ + dolomite assemblage, while quenching products of melts with Ca# < 20 will consist of $K_2Mg(CO_3)_2$ and magnesite (Figure 4a). Consequently, the following assemblages—$K_2Ca(CO_3)_2$ + dolomite, $K_2Ca(CO_3)_2$ + $K_2Mg(CO_3)_2$ + dolomite, and $K_2Mg(CO_3)_2$ + magnesite—can be expected among the daughter phases in carbonate-bearing melt inclusions in minerals from peridotite xenoliths.

## 5. Conclusions

The high-pressure alkaline carbonate melts were studied by Raman and SDD-EDS spectroscopy; according to the obtained data, we conclude the following:

(1) The high-pressure alkaline carbonate melts quench to the assemblages containing double carbonates, which are thermodynamically stable at a pressure of quenching. Thus, determination of the composition, structure, and Raman spectra of carbonate phases in

igneous minerals from kimberlites and lamproites, as well as in mantle xenoliths and natural diamonds, can help in determining the depth of their origin.

(2) The established fields of quench assemblages of the K-Ca-Mg carbonate melt at 3 and 6 GPa can be used for the reconstruction of the composition of carbonatitic melts entrapped by the diamonds and minerals of mantle xenoliths carried by kimberlites and, probably, magmatic minerals from kimberlites and lamproites.

(3) The K-Ca-Mg carbonate melt at 3 and 6 GPa quenches to carbonates thermodynamically stable at subsolidus conditions, but their quench assemblages may not coincide with subsolidus phase fields.

(4) The following assemblages—$K_2Ca(CO_3)_2$ + dolomite, $K_2Ca(CO_3)_2$ + $K_2Mg(CO_3)_2$ + dolomite, and $K_2Mg(CO_3)_2$ + magnesite—can be expected among the daughter phases in carbonate-bearing melt inclusions in minerals from peridotite xenoliths.

**Author Contributions:** Conceptualization, A.S.; methodology, A.S. and K.D.L.; validation, A.S.; formal analysis, A.V.A. and A.B.; investigation, A.V.A.; resources, K.D.L. and A.S.; data curation, A.V.A.; writing—original draft preparation, A.V.A.; writing—review and editing, A.V.A. and A.S.; visualization, A.V.A.; supervision, A.S.; project administration, A.V.A. and A.S.; funding acquisition, A.S. All authors have read and agreed to the published version of the manuscript.

**Funding:** This research was funded by Russian Foundation Basic Research, grant number 21-55-14001 and the state assignment of IGM SB RAS.

**Data Availability Statement:** The data presented in this study are openly available in [repository name e.g., FigShare] at [doi], reference number [reference number].

**Acknowledgments:** The SEM and EDX studies of experimental samples were performed in the Analytical Center for multielemental and isotope research SB RAS. We are grateful to N.S. Karmanov and A.T. Titov for their help in the analytical work.

**Conflicts of Interest:** The authors declare no conflict of interest. The funders had no role in the design of the study; in the collection, analyses, or interpretation of data; in the writing of the manuscript, and in the decision to publish the results.

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
