# Peer review of "Quench Products of K-Ca-Mg Carbonate Melt at 3 and 6 GPa: Implications for Carbonatite Inclusions in Mantle Minerals"

_minerals, doi:10.3390/min12091077_

Round 1

Reviewer 1 Report

The paper entitled "Quench Products of K-Cа-Mg Carbonate Melt at 3 and 6 GPa: Implications for Carbonatite Inclusions in Mantle Minerals" deals with the results of a series of experiments on synthetic alkali-carbonate melts aimed at establishing 1) if the K-Ca-Mg carbonates found within melt inclusions in diamonds, magmatic minerals from lamproites and minerals from mantle xenoliths carried by kimberlites can reflect the pressure at which the inclusions were trapped and 2) if the quench assemblages can reflect the bulk composition of the melt.

While the results of the experiments are well described and the conclusions are supported by the present study, the discussion should be deepened and the several paragraphs should be better linked together rather than leaving them as stand-alone units.

I suggest publication after corrections and revision of the manuscript by a mother-tongue speaker.

Here below my suggestions and comments following the line numbers of the manuscript:

1. Introduction

The introduction (lines 32-51) is very short and gaunt. The goal of the experiments is well explained (lines 46-51); however, the importance of carbonate inclusions in mantle materials for petrological studies and geodynamic reconstructions is only barely hinted at, and it would be necessary to deepen the topic. Why are alkali-rich carbonatite inclusions so important? What are the implications for the lithospheric dynamics?

2. Materials and Methods

- Lines 53-54: Please provide information  about the starting material (type of materials, for example powder, and their composition) used in the experimental runs.

- Lines 56-64: Please report precision and accuracy of the analytical method used to determine the composition of the run products, instead of simply referring to previous publications (22, 23).

3. Results

I am not an expert in experimental mineralogy, but I found the variation in the length of the experimental runs, spanning from 40 hours to only 15 minutes, a bit weird. Are 15-20 minutes sufficient to achieve equilibrium conditions?

-Line 84: Two Tables are cited but not present in the main text. No major element composition analyses of the phases have been carried out. I would appreciate the inclusion of EMPA analyses of some representative phases in the Tables or in the Supplementary Materials, if possible.

4.2. Implication for carbonatite inclusions

-Lines 212-220: The discussion here is not really clear and straightforward. Please re-phrase.

- Lines 222-232: the authors confirm the depth of provenance of a metasomatic ilmenite found by Giuliani et al. (2012) (5). However, this ilmenite was not "from a spinel harzburgite", as stated in the manuscript, but was from "ilmenite-rich domains emplaced in a spinel harzburgite wall rock". DU-1 xenolith studied by Giuliani et al. (2012) is a polymictic breccia and the ilmenite is probably not in equilibrium with the harzburgite, i.e. it has a distinct (metasomatic) origin.

4.4. Expected quench products of mantle melts from complex systems

- Lines 265-268: The statement is unclear, please re-phrase.

- Line 279: change“…melts with Ca# < 20 will be consists of…”, into “… melts with Ca# < 20 will consist of…”.

Figures

Figure 1: "Bu" – K2Ca(CO3)2 bütschliite is mentioned in the caption but it is not highlighted in the Figure.

Figure 4: Remove the list of authors for (1) and (5) in the caption.

Reviewer 2 Report

Review of Minerals-2022-1832276 by Arefiev et al.

Quench Products of K-Cа-Mg Carbonate Melt at 3 and 6 GPa: Implications for Carbonatite Inclusions in Mantle Minerals

This paper is proved by experiments that alkali-rich carbonate melts as inclusions can be used as markers reflecting the pressure of their entrapment and the assemblages of the quench products can reflect the bulk composition of the carbonate melt. The data is high quality and convincing. The origin of carbonate inclusions in mantle-derived rock is mysterious and important for our understanding of magmatic processes. Besides, the diversity of quench products of K-Cа-Mg carbonate melt is described from the perspective of mineralogy. Thus this study fits this well and worthy to be published on the journal, Minerals. However, this paper is still suffering some minor weaknesses that needs to be addressed. Hence, I suggest minor revision before it can be considered for publication.

Major comments:

1. Line 55. Are the initial material compositions you used same in all experiments? You'd better give full information of the starting material in the manuscript, for instance, the ratio of K2CO3, CaCO3, MgCO3 ratios, etc. In addition, please also list the experimental product phases in the manuscript instead of supplementary material as it is very important for readers to follow.

2. Line 85-86. You should explain what exactly you mean by using the K2#/Ca# ratio, e.g., why not use the K2#/Mg# ratio. When you calculate the ratio, it is best to indicate what is the composition unit of K2CO3, mol%?

3. In the first section of the discussion, how to use the canbonates as markers? Whether the two pressure conditions designed in this experiment span too wide? If there are more relevant experiment with pressures ranging from 3 GPa to 6 GPa that support your opinion, please list them.

4. During the processes of magma ascending, the pressure is decreasing, which could affect the stability of carbonate minerals. Hence, have you considered the kinetic process due to decompression?

There are many typos and errors in the text. Please check carefully before resubmission.

Round 2

Reviewer 3 Report

-
